

# Desiccation resistance: effect of cuticular hydrocarbons and water content in *Drosophila melanogaster* adults

Jean-Francois Ferveur[1], Jérôme Cortot[1], Karen Rihani[1], Matthew Cobb[2] and Claude Everaerts[1]

[1] Centre des Sciences du Goût et de l'Alimentation, Agrosup-UMR 6265 CNRS, UMR 1324 INRA, Université de Bourgogne, Dijon, France
[2] School of Biological Sciences, University of Manchester, Manchester, United Kingdom

## ABSTRACT

**Background**. The insect cuticle covers the whole body and all appendages and has bi-directionnal selective permeability: it protects against environmental stress and pathogen infection and also helps to reduce water loss. The adult cuticle is often associated with a superficial layer of fatty acid-derived molecules such as waxes and long chain hydrocarbons that prevent rapid dehydration. The waterproofing properties of cuticular hydrocarbons (CHs) depend on their chain length and desaturation number. *Drosophila* CH biosynthesis involves an enzymatic pathway including several elongase and desaturase enzymes.

**Methods**. The link between desiccation resistance and CH profile remains unclear, so we tested (1) experimentally selected desiccation-resistant lines, (2) transgenic flies with altered desaturase expression and (3) natural and laboratory-induced CH variants. We also explored the possible relationship between desiccation resistance, relative water content and fecundity in females.

**Results**. We found that increased desiccation resistance is linked with the increased proportion of desaturated CHs, but not with their total amount. Experimentally-induced desiccation resistance and CH variation both remained stable after many generations without selection. Conversely, flies with a higher water content and a lower proportion of desaturated CHs showed reduced desiccation resistance. This was also the case in flies with defective desaturase expression in the fat body.

**Discussion**. We conclude that rapidly acquired desiccation resistance, depending on both CH profile and water content, can remain stable without selection in a humid environment. These three phenotypes, which might be expected to show a simple relationship, turn out to have complex physiological and genetic links.

## INTRODUCTION

The resistant yet flexible outer layers of the insect exoskeleton, generally known as the cuticle, are involved in many vital functions and possess great structural, mechanical, and chemical complexity (*Locke, 1966*). The cuticle constrains the animal's shape, serves as an attachment point for muscles, and is the basis for a variety of specialized organs such as

Corresponding author
Jean-Francois Ferveur, jean-francois.ferveur@u-bourgogne.fr

sensory hairs, respiratory trachea, feeding and copulatory structures (*Demerec, 1950*). It also provides protection against physical and chemical environmental stressors such as desiccation. The cuticle not only limits desiccation (*Wigglesworth, 1945*), it also permits selective exchanges between the organism and the outer world (*Moussian, 2010*).

The soft and unpigmented cuticle of young adult insects undergoes a rapid process of melanization (*Andersen, 2010*) during which stabilized lipids, produced by the oenocytes and fat body, combine with proteins and stiffen the cuticle before sclerotization, which constitutes the final developmental phase (*Wigglesworth, 1988*). Most of our knowledge of the biochemical and genetic parthways involved in these processes comes from *Drosophila*, where they depend on the activity of the neurohormone, bursicon (*Honegger, Dewey & Ewer, 2008*), coupled with its rickets receptor (*rk*; *Harwood et al., 2013*), as well as on lipid production. A large part of the lipids contained within or secreted onto the epicuticle are long-chain hydrocarbons, generally known as cuticular hydrocarbons (CHs), the production of which depends on the *Cyp4g1* gene (*Qiu et al., 2012*), and on several genes coding for an elongase (*Chertemps et al., 2007*; *Howard & Blomquist, 2005*) and for substrate-specific lipid desaturases (*desat1, desat2*; *Bousquet et al., 2012*; *Dallerac et al., 2000*; *Flaven-Pouchon et al., 2016*). In *Anopheles gambiae*, a *Cyp4g1* ortholog, associated with insecticide resistance, catalyzes CH production (*Balabanidou et al., 2016*).

The waxes, lipids and CHs found on the epicuticle appear to have several functions. They act as a barrier against pathogenic microorganisms and insecticides (*Balabanidou et al., 2016*; *Da Silva et al., 2015*; *Goebiowski et al., 2008*), they reduce the heat load by reflecting solar radiation and they deter predators (*Eigenbrode & Espelie, 2003*; *Hadley, 1994*). They also help to limit transpiration (*Gibbs, 1998*; *Hadley, 1994*). Although the adult CH profile shows seasonal and environmental variation (*Gibbs & Pomonis, 1995*; *Howard & Blomquist, 2005*; *Savarit & Ferveur, 2002*; *Toolson & Hadley, 1979*) some of its components serve as sex- and/or species-specific pheromones (*Ferveur, 2005*; *Jallon, 1984*; *Savarit et al., 1999*). The *desat1* gene affects both the production and the reception of *Drosophila melanogaster* sex pheromones, while it has been suggested that *desat2* is involved in speciation (*Fang, Takahashi & Wu, 2002*). *desat1* is also expressed in tissues involved in water balance, in particular the Malpighian tubules (*Bousquet & Ferveur, 2012*; *Bousquet et al., 2012*; *Dow, 2009*).

Several studies of *D. melanogaster* have used artificial selection to enhance desiccation resistance by altering water loss, osmotic regulation and CH content. There is a clear link between desiccation resistance and water retention, dry mass and ionic content (*Folk & Bradley, 2004*; *Gibbs, Chippindale & Rose, 1997*), but the role of CH profile in preventing water loss in such selection experiments remains unclear (*Gibbs, Chippindale & Rose, 1997*; *Gibbs & Rajpurohit, 2010*). Selected lines showed no differences in the quantity of CHs, but sex differences in CH chain length were linked to variation in resistance—faced with a desiccation challenge, females survived longer than males (*Gibbs, Chippindale & Rose, 1997*; *Qiu et al., 2012*).

To reveal the link between desiccation resistance and CH profile in *D.melanogaster* flies, we carried out a three-part investigation using (i) experimentally selected lines, (ii) *desat1* transgenics and (iii) a *desat2* natural variant and *rk* mutants.

## MATERIALS AND METHODS

### Flies

*D. melanogaster* were raised on yeast/cornmeal/agar medium and kept at $24 \pm 0.5°$ with $65 \pm 5\%$ humidity on a 12 L:12 D cycle (subjective day $= 8:00$ am to 8:00 pm). Flies were isolated under light $CO_2$ anaesthesia either 0–4 h (for virgin females) or less than 24 h (for all other flies) after eclosion. Male and females were held separately in fresh glass food vials in groups of 10 flies until the day of the experiment (4–5 days old, unless specified). Same-sex flies were then transferred to an empty glass vial to obtain groups of $50 \pm 5$ individuals.

We tested two wild-type stocks, Dijon2000 (Di2) and Zimbabwe30 (Z30), which were collected in France and Zimbabwe respectively (*Grillet et al., 2012*; *Houot, Bousquet & Ferveur, 2010*). We also used the Di2/$w^{1118}$ line (Di2w), derived from the Di2 strain and introgressed into the genome of the $w^{1118}$ strain over five repeated backcross generations.

We used several *desat1*-Gal4 transgenic drivers built with putative *desat1* regulatory regions (PRRs) fused with Gal4. PRR(RA)-Gal4 is expressed in the wing margin and in the brain as well as in other tissues; PRR(RC)-Gal4 is exclusively expressed in the fat body; PRR(RE)-Gal4 is expressed in the oenocytes; PRR(RB) is expressed in the Malpighi tubules and midgut; PRR(RD) is expressed in a vaginal moon-shaped structure in the female and in the male ejaculatory bulb; PRR(RDiO)-Gal4 is expressed in neural tissues involved in the discrimination of sex pheromones by male flies (*Bousquet et al., 2012*). We also used a Gal4 transgene containing the complete *desat1* sequence fused to GAL4 (6908 bp; *6908*-Gal4; (*Bousquet et al., 2016*)). Gal4 drivers were used to target the UAS-*desat1*-RNAi transgene (IR; VDRC #33338; This UAS-RNAi line was chosen from three alternative lines for its clear effect on both the hydrocarbon and behavioral phenotypes tested here; *Bousquet et al., 2012*; *Houot et al., 2017*); this allowed us to down-regulate *desat1* expression in the tissues where Gal4 is expressed ([IR × driver-Gal4]; (*Houot et al., 2017*)). We also controlled for the effect of each driver-Gal4 in the Di2 background ([Di2 × driver-Gal4]). To homogenize genetic backgrounds, all UAS and Gal4 transgenes were isogenized in the genetic background of the Di2/$w^{1118}$ strain prior to testing.

Two *rickets* alleles were used, *rk[1]* and *rk[4]*, which contain different mutations that create premature stop codons, and are considered null alleles (*Baker & Truman, 2002*). We carried out reciprocal crosses between virgin adults carrying the *rk[1]* or the *rk[4]* mutation, both of which were maintained in a heterozygous state by the [SM2, CyO] balancer chromosome. From this cross, we obtained *rk[1]*/*rk[4]* heteroallelic mutant adults and *rk[1]*/CyO and *rk[4]*/CyO control flies. To control the effect of the SM2, CyO balancer chromosome, we mated Di2w females with either *rk[1]*/CyO or *rk[4]*/CyO males and collected all CyO flies (i.e., those not carrying the *rk* mutation; CyOw).

### Experimental selection

Groups of same-sex flies ($n = 50 \pm 5$) of a given age were kept in glass vials (Fig. 1A). Six or seven vials were kept in a transparent plastic box, which was partly filled with silica gel to reduce the humidity to 20% RH and was secured with transparent adhesive tape. We simultaneously tested three or four boxes placed on a hot plate at $25.0 \pm 0.2°$ (StörkTronic,
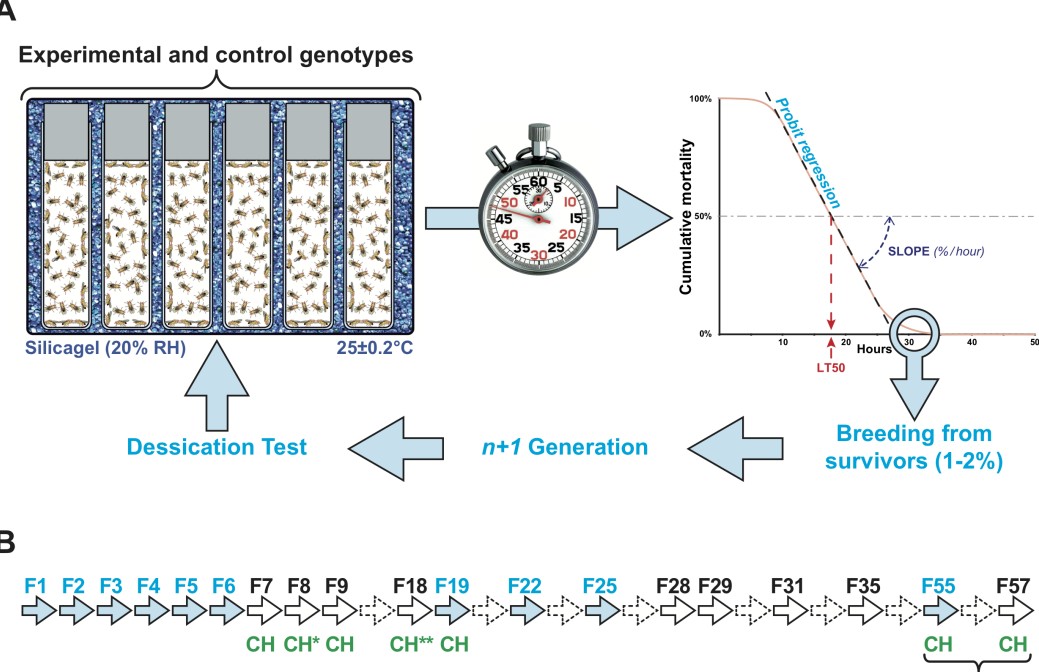

**A**

Experimental and control genotypes

Silicagel (20% RH)    25±0.2°C

Cumulative mortality

Probit regression

SLOPE *(% / hour)*

LT50

Dessication Test ← *n+1 Generation* ← Breeding from survivors (1-2%)

**B**

F1 F2 F3 F4 F5 F6 F7 F8 F9    F18 F19    F22    F25    F28 F29    F31    F35    F55    F57

CH CH* CH    CH** CH                                                    CH        CH

W                                                                        W
                                                                         Fec

**Figure 1 Experimental selection of desiccation resistance line.** (A) Flies were kept in single sex groups of approximately 50 individuals in empty glass vials. Six or seven of these glass vials were packed inside an airtight transparent plastic box that was seeded with a layer of silicagel crystals to maintain a low relative humidity (20 ± 1%). The box was placed on a hot plate at 25 ± 0.2°. Four boxes were simultaneously tested. Every two hours, the number of dead flies was counted, providing a measure of survival over time. This allowed us to estimate when 50% flies died (lethality time 50% = LT50), using logistic regression. The slope of the lethality curve was also determined to evaluate the relative lethality per hour. At the end of each experiment, the few surviving flies (maximum 1–2% of all flies) were transferred into fresh food vials and mated with siblings to produce the next generation. (B) Arrows indicate the generations of the experiment. Plain arrows indicate the generations during which experimental selection for desiccation (light blue) and/or phenotypic measurements (CH, cuticular hydrocarbons; W, weight; Fec, fecundity) were carried out. Dashed arrows indicate the generations during which no measurement or selection was carried out.

Präzitherm, Düsseldorf, Germany). Every hour or two hours, we counted the number of dead flies in each vial. In each box and in each experiment, we mixed vials containing the different genotypes. For each generation, we performed two or three series of tests that were subsequently pooled.

To experimentally select lines for desiccation resistance, the 1–2% flies showing the longest life span during the desiccation challenge were placed in a fresh food vial and allowed to mate, producing the next generation (Fig. 1B). This procedure, begun with the wild-type Di2 strain, was carried out over six successive generations (F1–F6) and was then sporadically carried out until F57. Of ten F1 lines, flies of the #7 line, which showed high desiccation resistance, were used to create six F2 lines called 77S, which were subject to subsequent selection. Between F2 and F6, we also tested flies resulting from the backcross between 77S females and Di2 males (77S × Di2) and the reciprocal backcross. After F6, no

selection was carried out on 77S lines except to create six 77S-Sel lines which were subject to selection; each of these derived from their respective 77S lines (e.g., 77S-Sel1 derived from the $77S_1$ line).

## Cuticular hydrocarbons

Five-day-old flies were frozen for 5 min at $-20$ °C and their cuticular hydrocarbons then individually extracted for 5 min at room temperature using 30 µl of a mixture of hexane and methylene chloride (50/50 by volume). The solution also contained 3.33 ng/µl of C26 (*n*-hexacosane) and 3.33 ng/µl of C30 (*n*-triacontane) as internal standards. Cuticular hydrocarbons were quantified by gas chromatography using a Varian CP3380 gas chromatograph fitted with a flame ionization detector, a CP Sil 5CB column (25 m × 0.25 mm internal diameter; 0.1 µm film thickness; Agilent), and a split–splitless injector (60 ml/min split-flow; valve opening 30 sec after injection) with helium as carrier gas (50 cm/s at 120 °C). The temperature program began at 120 °C, ramping at 10 °C/min to 140 °C, then ramping at 2 °C/min to 290 °C, and holding for 10 min. Individual CH profiles were determined by integration of 46 peak areas in males and females. This corresponded to all the peaks that could be consistently identified in all individuals (*Everaerts et al., 2010*). The chemical identity of the peaks was checked using gas chromatography—mass spectrometry equipped with a CP Sil 5CB column. The amount (ng/insect) of each component was calculated on the basis of the data obtained from the internal standards. We calculated the absolute amount of each group of CHs (alkene Q, alkane Q), the relative amount of each CH group (alkene %, alkane %) from the overall CH total ($\sum$CH) and their ratio (Desaturated:Linear = D:L). At least 10 flies were tested per condition.

## Water content

Groups of 10 live anaesthetized females were weighed on a precision balance ($\pm 10$ µg; Sartorius R160-P) to obtain their fresh weight. Each group of females was then kept for 24 h in an empty glass vial in a 37° dry incubator to allow complete desiccation. The dead, dry flies were then weighed to obtain the dry weight. The relative level of water in each group was estimated based on the fresh weight:dry weight ratio.

## Fecundity

Females and males were kept in groups of ten pairs until they were four days old. Females were then isolated (males were discarded) and the total number of male and female adult progeny was noted for seven days following the emergence of the first offspring. The sex ratio of the progeny was also noted.

## Statistics

All statistical analyses were performed using XLSTAT 2012 (*Addinsoft, 2012*). For each desiccation replicate, logistic regression was used to characterize the relationship between mortality and time by estimating the lethal time 50 (LT50) and the regression slope (*Robertson & Preisler, 1992*). Thereafter, for each generation an inter-line comparison for these two parameters was carried out either with a Kruskall-Wallis test with Conover-Iman multiple pairwise comparisons ($p = 0.05$, with a Bonferroni correction) or with a Mann–Whitney test, after excluding extreme outliers using Tukey's method (*Tukey, 1977*). The

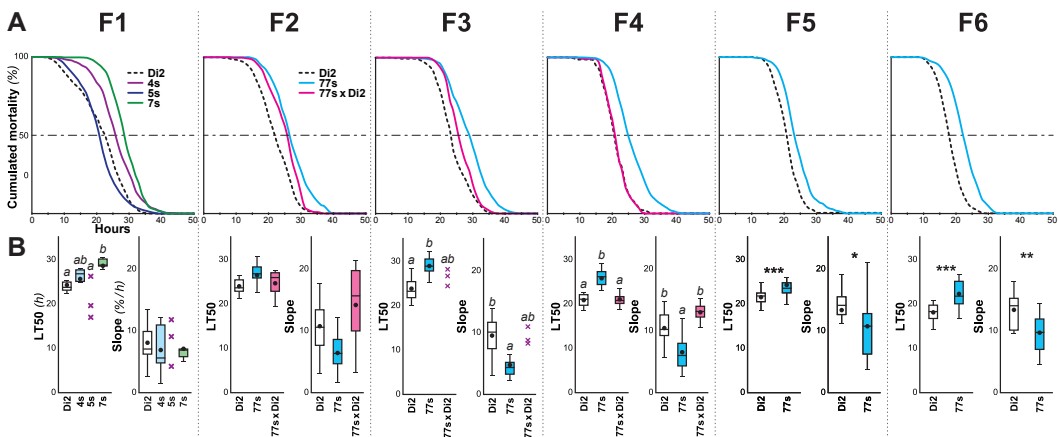

**Figure 2** **Survival in females selected for desiccation resistance over the first six generations.** Female flies were selected using the experimental procedure described in Fig. 1. (A) For each generation (F1 to F6), the curves represent the survival measured in various genotypes (dashed line, Di2 control line; cyan line, 77S selected lines pooled; magenta line, backcross between 77S females and unselected sibling males; at F1, three selected lines are shown). (B) At each generation, the two box-plots represent the LT50 and the lethality slope using colors similar to those of the corresponding genotypes. Data are shown as box plots representing the 50% median data (the small horizontal bar indicates the median value while the plain dot represents the mean). The whiskers shown below and above each box represent the first and third quartiles, respectively. Stars or different letters indicate significant differences. After excluding extreme outliers using Tukey's method, LT50 and slopes were tested using a Kruskall–Wallis test completed by a Conover-Iman multiple pairwise comparisons at level $p = 0.05$ (with a Bonferroni correction) or with a Mann–Whitney test. ***, $p < 0.001$; **, $p < 0.01$; *, $p < 0.05$. The absence of a letter or stars indicates that no significant difference was detected. $N = 5$–$17$ (except 5S line at F1 and 77s × Di2 line at F3 where $N = 3$). A similar selection procedure was carried out on males between the F1 and F6 generations (Fig. S2).

overall amount of CH ($\sum$CH), the Desaturated:Linear ratio (D:L), the fresh weight:dry weight ratio, the total number of adult progeny and the sex ratio were also compared using the same statistical tests.

## RESULTS

### Selection for desiccation resistance

Five- to seven-day-old flies were placed in groups in a relatively dry environment (20% RH) at $25 \pm 0.2\,°C$, and two measures of survivorship were taken: LT50 (time at which 50% flies were dead) and lethality slope (the steepness of this curve indicates the proportion of flies dying per hour; Fig. 1A). Males were significantly more affected by the dry conditions than females, as shown by a shorter LT50 and a steeper slope (KW$_{(5df)} = 31.74$, $p < 10^{-4}$; Fig. S1). Because of this sex difference, we subsequently focused on female resistance. No significant differences were found between virgin and mated females (see 'Materials and Methods'). Survivors of this initial desiccation challenge were allowed to mate, and a selection experiment on desiccation resistance was then undertaken, with eight replicate lines. After only one generation, significant resistance appeared in one line (#7; Fig. 2); we therefore focused on this line, crossing #7 flies to create six replicate selected lines, known

as $77S_{0-5}$. These lines were then used in our experiment on desiccation resistance; data from all six 77S lines were pooled at each generation for statistical analysis.

From F3 to F6, 77S females showed significantly increased desiccation resistance as compared to control Di2 flies, as shown by a higher LT50 ($p < 10^{-4}$–$10^{-3}$) and a reduced slope ($p$: 0.001–0.037). In order to explore the genetic control of these characters, female and male 77S flies were separately backcrossed to control Di2 flies, and their desiccation resistance was measured. Although both the LT50 and slope of the offspring of the [77S f × Di2 m] backcross were intermediate between control and 77S lines, flies produced by the reciprocal backcross [Di2 f × 77S m] were not significantly different from control ($p = ns$). The effect of selection on male flies was much less, if any (Fig. S2). These data suggest that the character(s) that have been selected for in the 77S flies are primarily transmitted through female flies.

After F6, systematic selection of the 77S lines was relaxed (Fig. 1B). Over 52 subsequent generations, desiccation resistance was sporadically tested in the $77S_{0-5}$ lines; we also re-selected females from each 77S line for one or two generations prior to these desiccation resistance tests and tested their progeny (77S-Sel; Fig. 3). Compared to control females, 77S females showed a significantly increased LT50 (median value ranges: 14.83–21.67 h and 18.77–24.45 h, respectively) and/or a shallower slope (median value ranges: 9–16 and 5–9 % lethalithy/h, respectively) that showed little consistent variation over time. Reintroduction of selection in the 77S-Sel lines had no effect—these flies were not significantly different from 77S flies that were reared under relaxed selection, indicating that the character(s) isolated in the selection procedure are at fixation in these lines.

### Effects on associated characters

Cuticular hydrocarbons (CH) have regularly been implicated in the evolution of desiccation resistance; we therefore measured CH profiles in F7–F9 $77S_{1-5}$ females and in F8 males, and sporadically thereafter between F18 and F59. Beside the absolute CH amount ($\sum CH$), we also determined the absolute (Q) and relative amounts (%) of desaturated CHs (alkenes) and of linear saturated CHs (alkanes) and their ratio (Desaturated:Linear = D:L). Although we observed both interline and intergenerational differences in the $77S_{0-5}$ flies (Fig. 4), all F9 females showed increased levels of alkene Q and most showed an increased alkene % ($77S_4$ was an exception). Compared to control Di2 females, most lines showed higher $\sum CH$ and D:L (Fig. S3—$77S_0$ was an exception). At F18, following 12 generations of relaxed selection, only $77S_1$ and $77S_2$ females showed increased D:L, while at F19 only $77S_2$ females showed increased D:L (Fig. 5).

At F55, we tested $77S_1$–$77S_5$ flies (the $77S_0$ line was lost between F35 and F55), all of which showed a significantly increased D:L, due to their higher alkene levels (Q and %) and lower alkene Q (Fig. 5 and Fig. S4). In all 77S lines, $\sum CH$ was significantly higher than in Di2 females. At F57, 77S females were compared with females that had been re-selected at F55; 77S-Sel—Fig. 5). D:L increased in three 77S lines ($77S_1$–$77S_3$), but not in 77S-Sel lines. All 77S lines showed increased $\sum CH$, while only one of the 77S-Sel lines showed such an effect. These differences between 77S and 77S-Sel lines contrast with the results of the desiccation resistance experiments, where there were no significant differences between

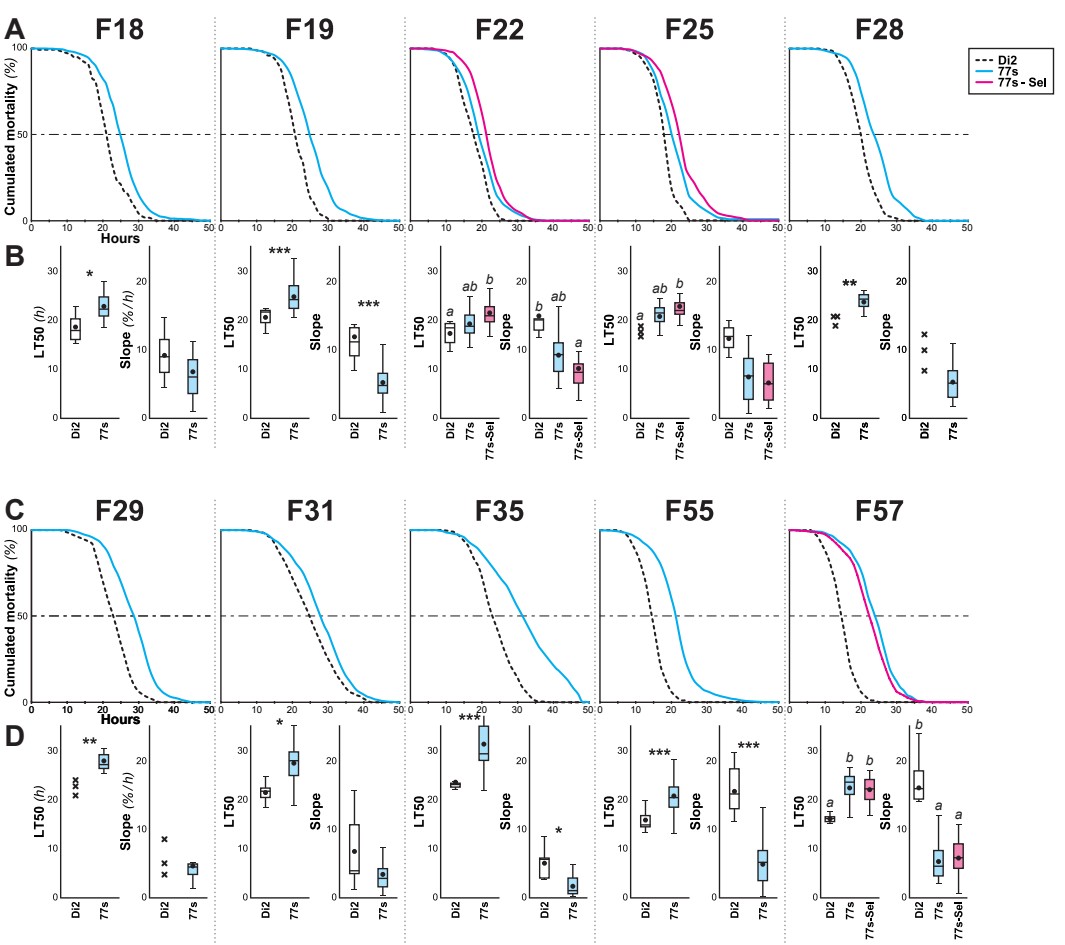

**Figure 3** **Survival in females of selected lines between F18 and F57.** Female flies were selected at the indicated generations using the procedure described in Figs. 1 and 2. More precisely, we show the survival curve (A, C) and the LT50 with the slope (B, D) for the generations F18–F28 (A, B) and F29–F57 (C, D). $N = 5$–22 (except Di2 line at F18: $N = 4$, and at F25, 28 & 29: $N = 3$). For parameters and statistics, see Fig. 2 legend.

these sets of lines, and suggest that CH composition and desiccation resistance are not identical.

In order to explore the link between CH composition and desiccation resistance, we measured the fresh and dry weight of flies (either freshly killed, or desiccated, respectively) from these lines at F19 and F57/59. Fresh and dry weight can be considered as indirect measures of cuticular surface and their ratio reflects the water retention ability of a particular strain. In F19 females, the Fresh:Dry weight ratio was significantly higher in lines $77S_1$–$77S_3$ than in Di2 flies (Fig. 6A). However in F57/F59 females, no difference was detected between Di2 and 77S females (Fig. 6B).

Finally, to confirm that we had not inadvertently selected for changes in sex ratio or number of eggs laid by the 77S females, we counted the total number of adult progeny left by single 77S, 77S-Sel and control females, and calculated their sex-ratio (female:male).

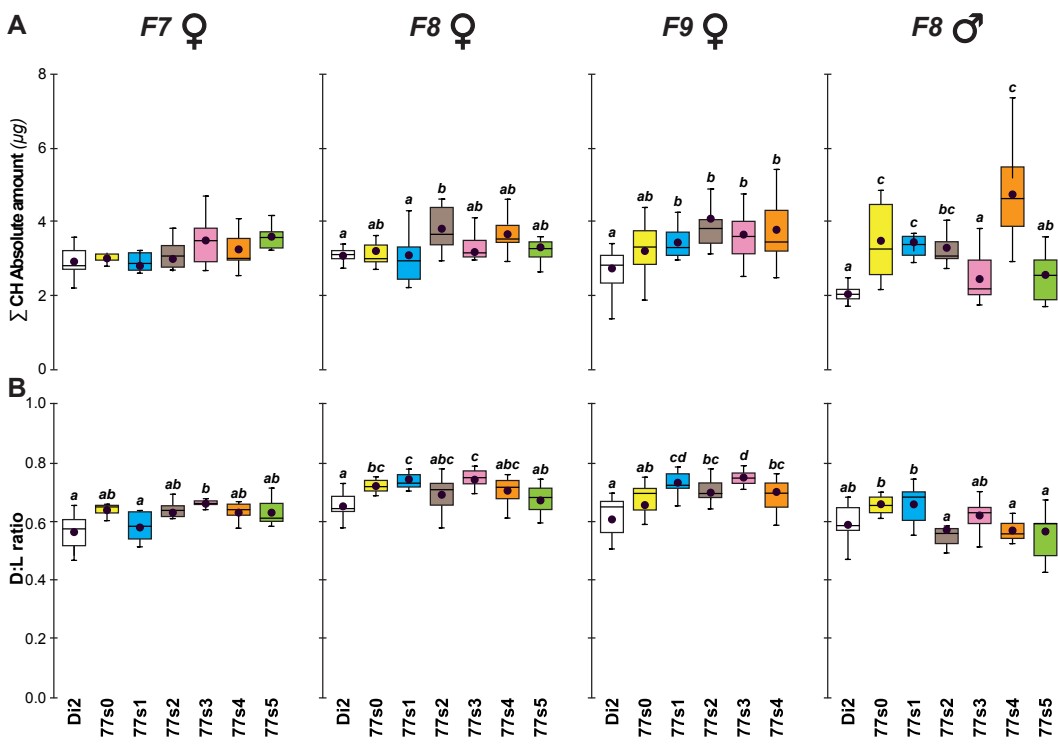

**Figure 4** **Principal cuticular hydrocarbons in flies of selected lines following relaxation of selection.**
Cuticular hydrocarbon levels (CHs) were measured in F7, F8 and F9 females and in F8 males separately in
the six 77S lines ($77S_0$–$77S_5$) experimentally selected for desiccation resistance (F1–F6; see Fig. 2). Here,
we show the total absolute amount of CHs ($\sum$CH in $\mu$g, A) and the ratio of Desaturated: linear saturated
CHs (D:L ratio; B) This ratio was calculated using the formula ([D—L]/[D + L]). $N$ = 5–20 for females
and $N$ = 9–14 for males. For statistics, see Fig. 2 legend. We also determined the absolute (Q) and relative
(%) amounts of desaturated CHs (alkenes) and of linear saturated CHs (alkanes) (Fig. S3).

There were no overall differences between these groups (Fig. S5). Only $77S_3$ females
showed a significant variation by producing more progeny than Di2 females (116 and 66,
respectively) and more males (64 and 32, respectively), but this did not affect the sex ratio.

## Genetic control—*desat1*

A major gene involved in CH synthesis is *desat1*, which controls a vital desaturation
step in the hydrocarbon biosynthetic pathway. To explore the role of *desat1* in desiccation
resistance, we downregulated this gene in subsets of tissues and measured the consequences
for desiccation resistance and associated characters. Driver-Gal4 lines, made either with
each putative *desat1* regulatory region (PRR-Gal4) or the complete *desat1* regulatory region
(6908-Gal4), were used to drive the expression of the UAS-*desat1* RNAi transgenic reporter
line (IR). This allowed us to downregulate *desat1* expression either in Gal4-targeted tissues
([PRR-Gal4 × IR]) or in all *desat1*-expressing tissues ([6908-Gal4 × IR]). Under the
desiccation conditions used in the selection experiment, [RC-Gal4 × IR] and [6908-Gal4
× IR] females showed reduced desiccation resistance, as shown by a reduced LT50 (Fig. 7A),
but no difference in the rate at which flies died (as measured by the slope; Fig. 7B). This

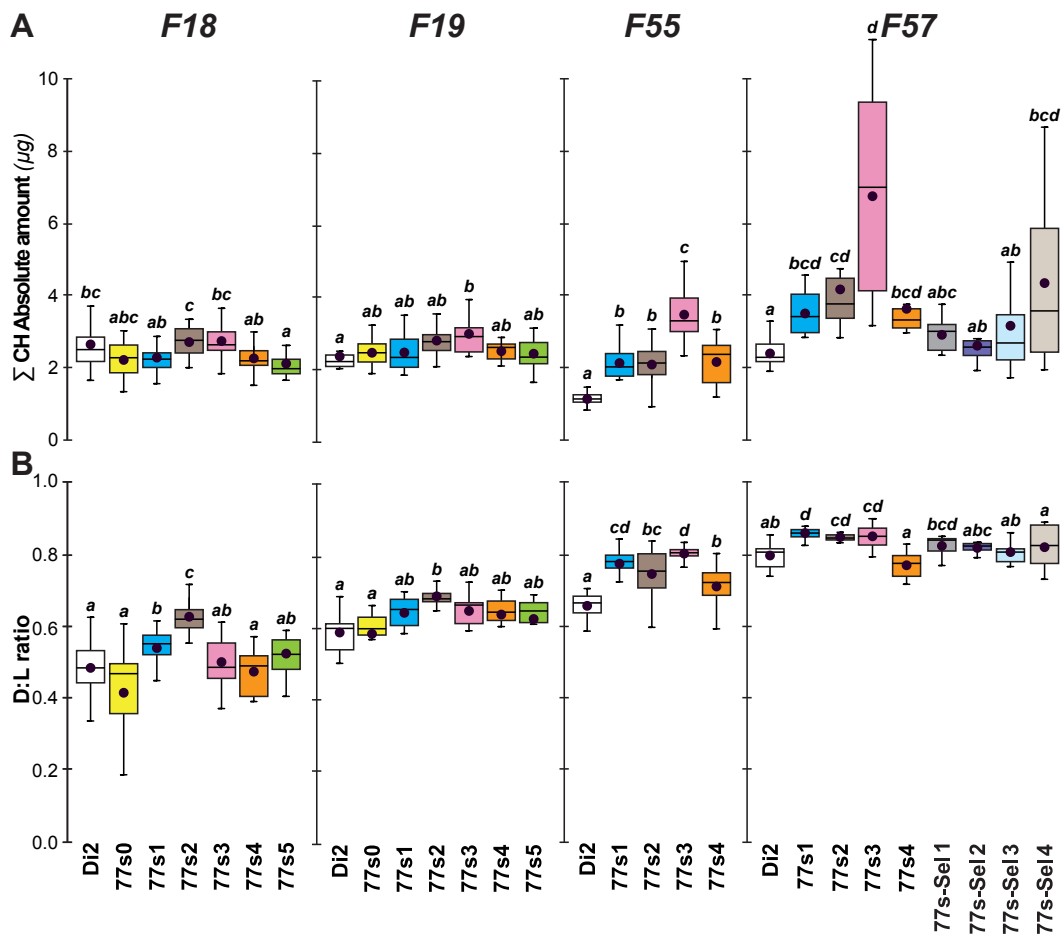

**Figure 5  Principal cuticular hydrocarbon levels in females of selected lines between F18 and F57.**
$\sum$CH (A) and D:L ratio (B) were measured in F18, F19, F55 and F57 females. As well as control unselected Di2 females, six 77S lines ($77S_0$–$77S_5$) were tested in F18 and F19; only four of these lines ($77S_1$–$77S_4$) survived to F55 and F57. The 77S-Sel lines (77S-Sel1–77S-Sel4) that were tested at F57 were the offspring of F55 reselected females from their respective 77S lines (e.g., $77S_1$ females yielded the 77S-Sel1 line). $N = 7$–38. For more information on parameters, lines and statistics, see legends of Figs. 2 and 4. The absolute (Q) and relative (%) amounts of desaturated CHs (alkenes) and of linear saturated CHs (alkanes) determined in these flies are shown in Fig. S4.

suggests that expression of *desat1*, in particular in the fat body, is required for normal desiccation resistance. Control genotypes were tested simultaneously and showed no effects (Figs. 7C and 7D).

To confirm that manipulation of *desat1* had altered the CH profiles, the CH levels of [IR × driver-Gal4] females in two generations ($F_A$ and $F_B$) were compared to simultaneously-raised controls (Di2; Di2w; [Di2 × IR]; [driver-Gal4 × Di2]; Fig. 8). Despite slight quantitative variations between the two generations Figs. 8A and 8B), $F_A$ and $F_B$ flies showed similar effects. In particular, [RE-Gal4 × IR], [6908-Gal4 × IR] and to a lesser extent [RC-Gal4 × IR] flies produced lower alkene levels (Q and %; Fig. S6). [RE-Gal4 × IR] and [6908-Gal4 × IR] flies also showed increased alkane levels (Q and %) and

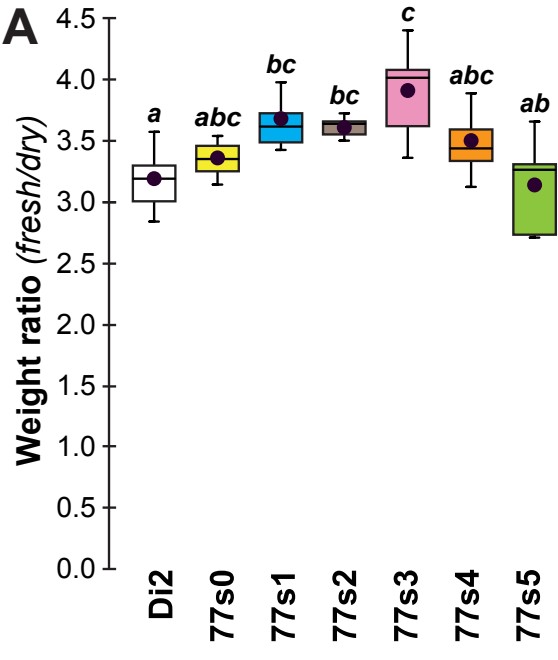

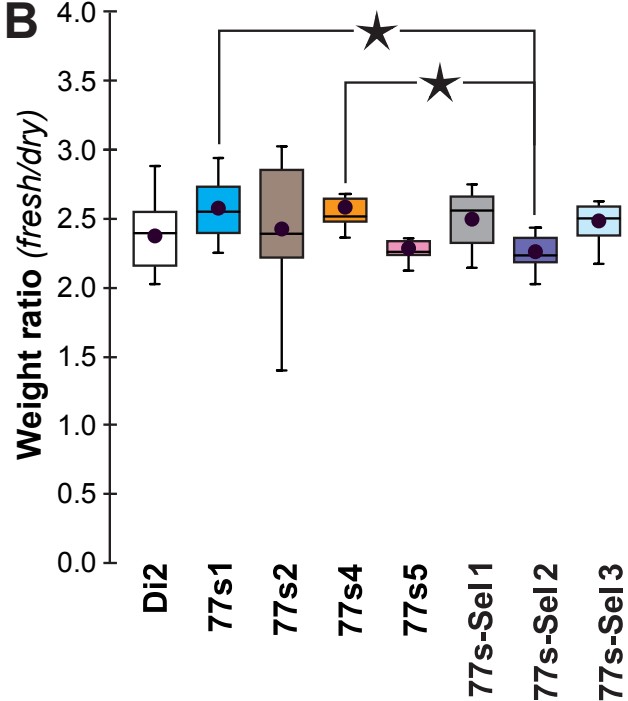

**Figure 6** **Fresh:dry weight ratio in females of selected lines.** Groups of 10 freshly killed females were weighed (fresh weight) and after 24 h desiccation were weighed again (dry weight). The fresh:dry weight ratio of each group was calculated. Females were weighed at F19 (A) and F57 (B). $N = 6$–20. For more information on genotypes and statistics, see legends to Figs. 2, 4 and 5.

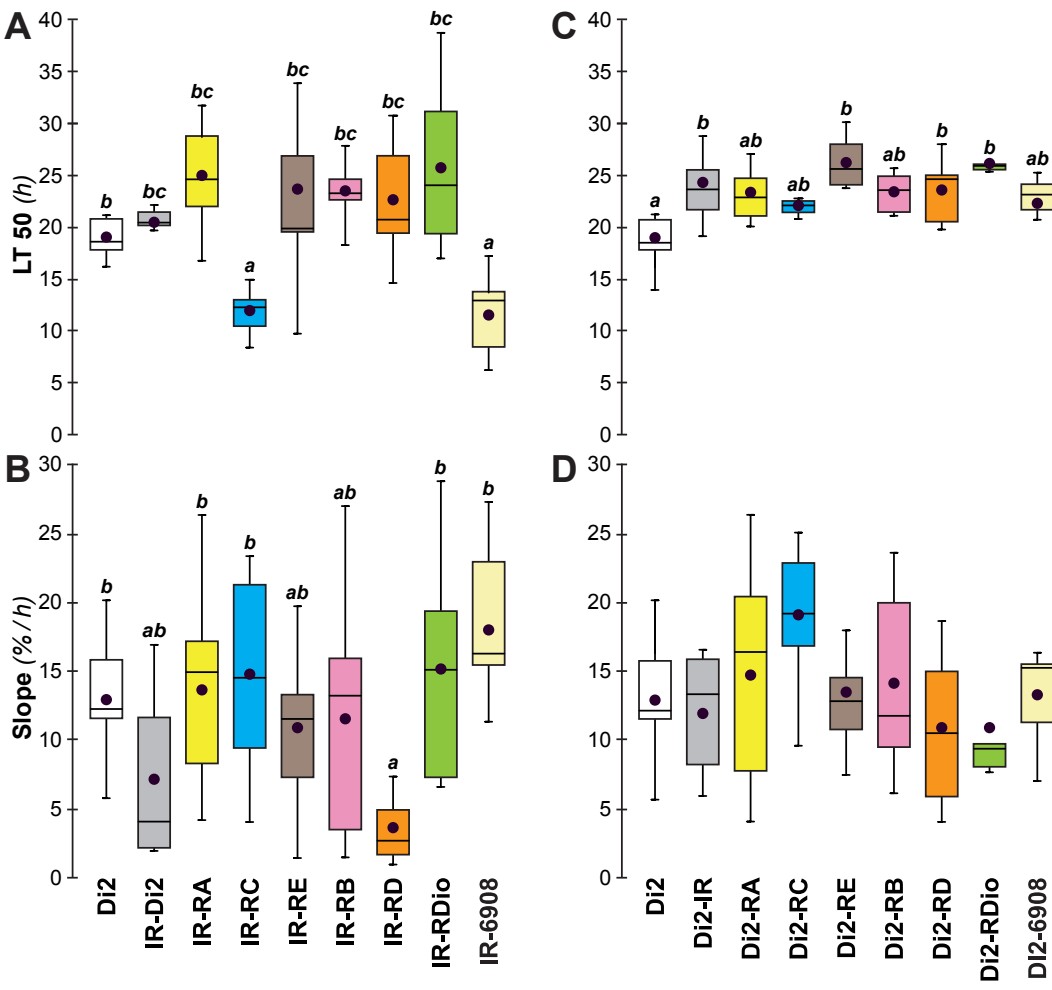

**Figure 7  Desiccation resistance in various *desat1* transgenic females.** To test the effect of *desat1* knockdown expression in various *desat1*-expressing tissues, we used the female progeny of matings between transgenic females carrying the UAS-*desat1*-IR transgene (IR) and transgenic males either carrying each *desat1* putative regulatory region fused with Gal4 (PRR-Gal4) corresponding to each *desat1* transcript (RA, RC, RE, RB, RD, RDiO), or the complete *desat1* regulatory region (6908 bp = 6908). Di2 control females and female progeny resulting of matings between IR females and Di2 males were also tested (Di2; Di2-IR; left box plots). The LT50 (A) and the lethality slope (B) of all these genotypes were determined. To control for the effect of each *desat1* PRR-Gal4 transgene on the LT50 (C) and lethality slope (D), we used flies from matings between Di2 females and PRR-Gal4 males, alongside Di2-IR and Di2 control females. $N = 5$–13. For more information on parameters and statistics, see Fig. 2 legend.

lower D:L compared to controls. Comparison of Fresh:Dry weight revealed a significant increase in [RC-Gal4 × IR] females compared to control lines (Fig. 9), paralleling the effects on CH levels shown by this line. Although [6908-Gal4 × IR] flies also appeared to show an increased ratio, this was not significant compared to [Di2 × IR] controls. No differences in average fecundity were found when $F_A$ and $F_B$ flies were compared with controls (Fig. S7).

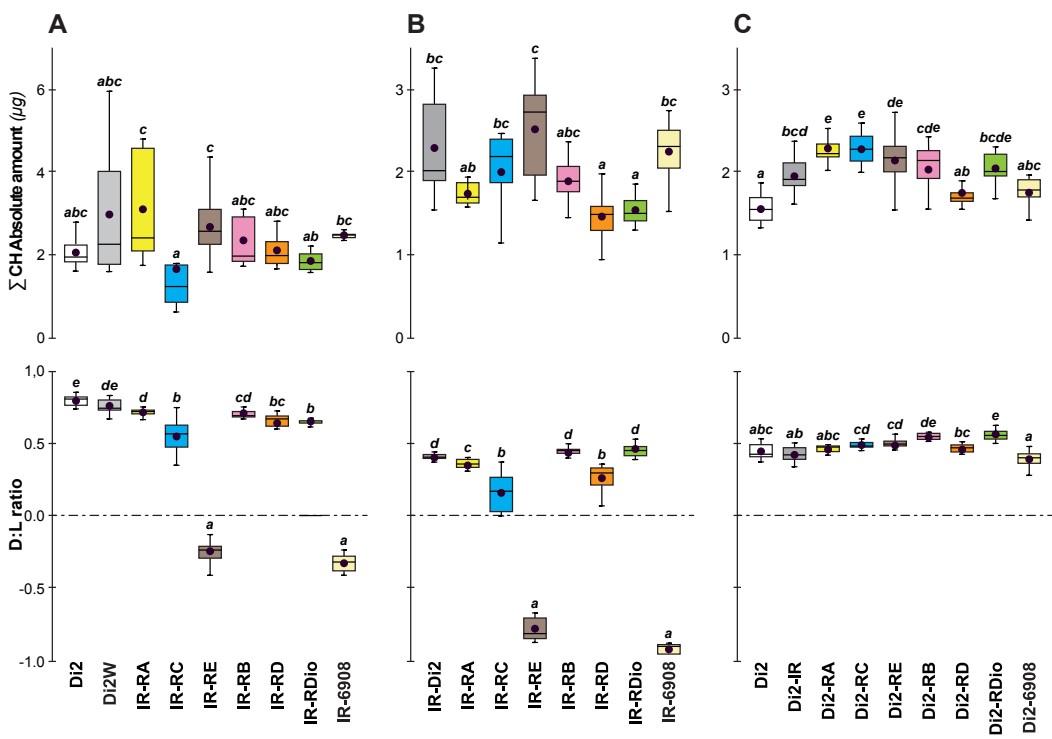

**Figure 8** **Principal cuticular hydrocarbons in various *desat1* transgenic females.** CH levels were measured in all transgenic and control female flies tested for desiccation resistance (see Fig. 7). Transgenic females combining a maternal IR transgene with a paternal PRR-Gal4 or 6908 transgene were tested at both F55 (A) and F57 (B). Di2, Di2/*w* and Di2-IR control females were tested (left box-plots) either at F55 (A) or at F57 (B). Control genotypes carrying a paternal copy of each PRR-Gal4 transgene or of the 6908 transgene combined with a maternal Di2 genome were also tested (C). Alongside these control genotypes, we also tested the effect of the IR transgene in the Di2 background (second box-plot from the left). $N = 7$–16. For more information on CHs, genotypes and statistics, see legends to Figs. 4 and 7.

## Genetic control—natural and lab-induced variants

To further explore genetic control of the link between desiccation resistance and CH profile, we studied naturally-occurring and laboratory-induced variants. In Zimbabwe (Z30) flies, the female-specific alkene isomer 5,9-heptacosadiene (5,9 HD) largely replaces 7,11 HD which is abundant in the other strains studied here (*Flaven-Pouchon et al., 2016*; *Grillet et al., 2012*). The desiccation resistance shown by these females was not significantly different from Di2 flies. $rk^{1/4}$ mutant females produce high absolute amounts of CH ($\sum$CH, alkene and alkane) but control-like alkene and alkane % and D:L; these mutants showed significantly lower resistance than control females (lower LT50, but no slope variation) (Fig. 10). However, *rk*/CyO females showed significantly greater resistance than controls, with a highly increased LT50 and a strongly decreased slope. Increased resistance in *rk*/CyO females was not related to the presence of the CyO balancer –CyO*w* females did not show the same effect. Finally, there was a slightly increased resistance (LT50 and slope) shown by Di2*w* females compared to Di2 females.

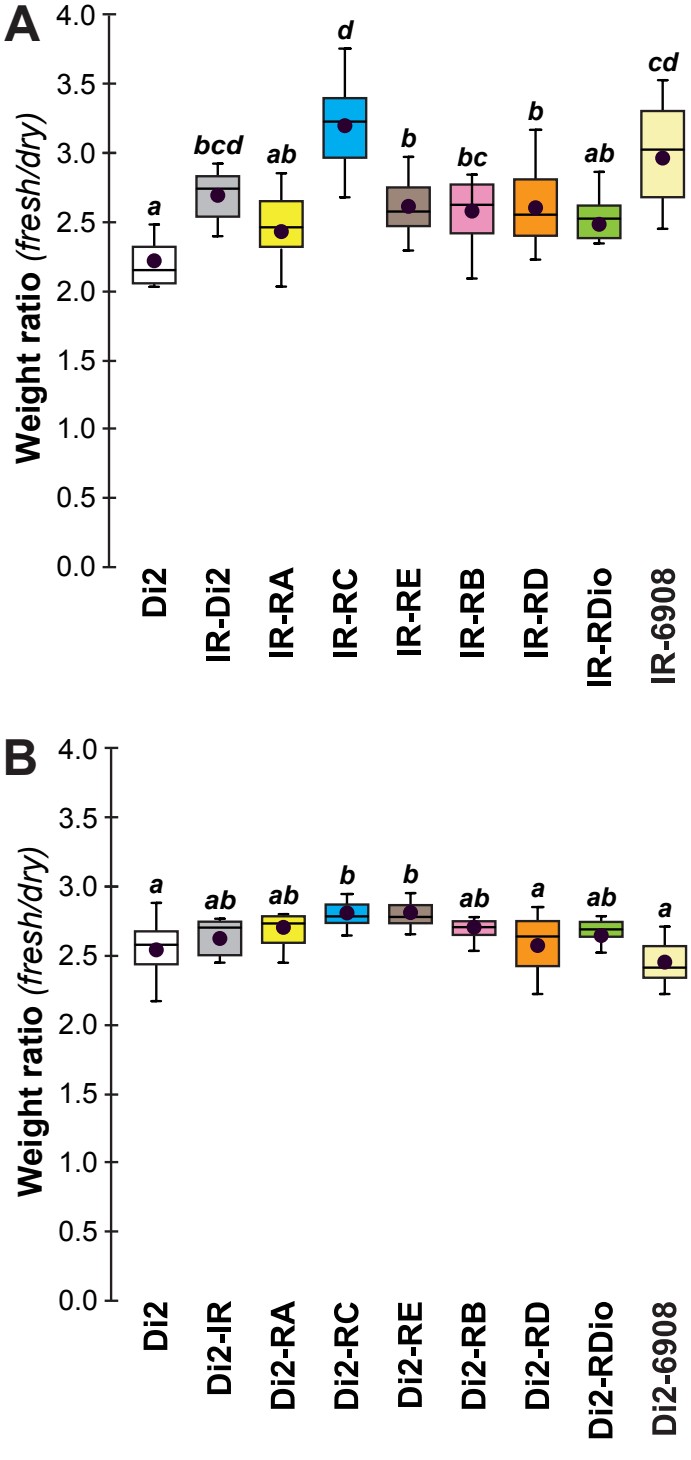

**Figure 9** **Fresh:dry weight ratio in various *desat1* transgenic females.** The fresh: dry weight ratio were analyzed in transgenic (A) and in control genotypes (B) $N = 5–15$. For more information on genotypes and statistics, see legends to Figs. 2 and 7.

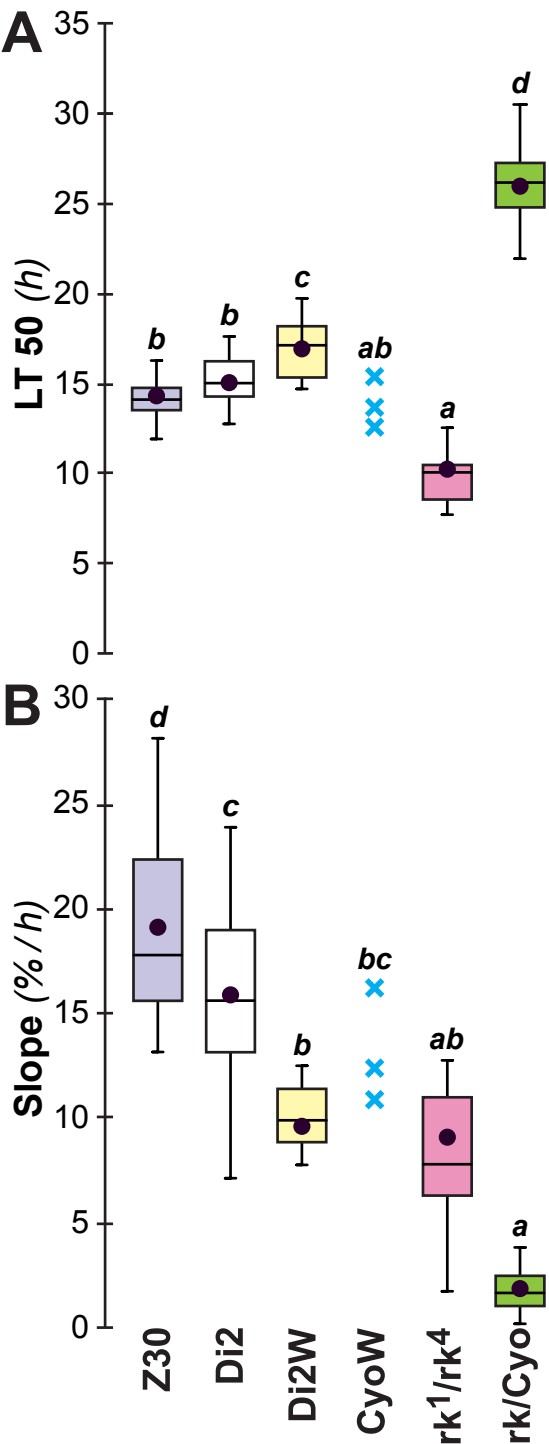

**Figure 10 Survival to desiccation in various cuticular hydrocarbon variant females.** The LT50 (A) and lethality slope (B) of various CH variants was measured. We also assayed the resistance of flies from these lines: control Di2, Di2/w (combining the Di2 genome with the $w^{1118}$ white-eye mutation), Zimbabwe 30 line (Z30), $ricket^{1/4}$ double trans-heterozygote mutant ($rk^1/rk^4$), $rk$ heterozygotes carrying the SM2 balancer ($rk/CyO = rk^1/CyO$ and $rk^4/CyO$), and Di2/w flies carrying the SM2 balancer (CyO/w). $N = 10$–30 (except for CyO/w line: $N = 3$). For more information on statistics, see legend to Fig. 2.

## DISCUSSION

Desiccation is a major physiological challenge faced by terrestrial arthropods, and both experimental and theoretical arguments have been deployed to suggest that there is a physiological and evolutionary link between desiccation resistance and the presence of certain cuticular hydrocarbons, which act as a waxy outer layer restricting permeability (*Gibbs, 1998*; *Qiu et al., 2012*; *Toolson & Hadley, 1979*). Using a range of experimentation approaches, we were able to reveal genetic variation for desiccation resistance in our laboratory strain of *D. melanogaster*, as shown by the appearance of a desiccation-resistant strain (#7) after one generation, and then a rapid response to selection for desiccation resistance in multiple replicates of that line which was maintained over dozens of generations despite the subsequent relaxation of selection. Furthermore, we found correlated changes in cuticular hydrocarbon composition in these replicate lines, in particular the overall amount of CH and the D:L ratio, together with water retention ability. We suggest that these were causally linked—altered cuticular hydrocarbon composition enabled the selected lines to retain water and resist desiccation. These findings suggest that genetic variability for desiccation resistance exists even in laboratory lines (the Di2 line was captured in 2000); variability in wild populations is presumably greater, and may help explain the global success of this species.

The detection of significant desiccation resistance in line #7 after a single generation enabled us to select for that character over subsequent generations, significantly altering both LT50 and the lethality slope. From the outset, we found that females were significantly more responsive to selection than males, which remained significantly more affected by the desiccation challenge than females. This led us to focus on the female phenotype for the rest of the experiment. Two possible explanations for this sex difference are that the genes we were selecting were sex-limited, expressed only in females, linked to qualitative and quantitative sex differences in CHs (see below), or that the effect was simply due to size—female *Drosophila* are larger than males, with a lower surface area:volume ratio, which in turn would reduce the effect of desiccation.

Although selection was relaxed after F6, the resistance character we had selected in the #7 line was stable for over 50 generations, with no significant effect of the reintroduction of selection, indicating that the character(s) involved were at fixation. The fact that these characters were detected after one generation and became fixed after a further five generations of selection suggests that our selection protocol and the available genetic variation in the parent population were well matched and the strongest possible phenotypes were rapidly selected through the available genetic variation.

At first glance it could be thought that the main effects we observed—increased desiccation resistance, increased quantities of CH, and increased water content—were all produced by an increase in the size of female flies, which would also increase the quantity of CH by increasing the surface area. Such an increase in size would lead to a decreased surface area: volume ratio, thereby increasing water retention at least at some time points. However, there were no evident size increases in the selected females—certainly nothing approaching the difference in size between males and females, which is visible to

the naked eye—and we consider that any potential microscopic changes in size would be unlikely to produce the significant differences in survival we observed in the selected lines.

Insight into the link between the effects on CH levels, water content and desiccation resistance can be found by inspecting the genetic mapping of desiccation-related phenotypes in *desat1* transgenic flies. The selective knock-down of *desat1* expression in the female fat body simultaneously induced a decreased desiccation resistance and a markedly increased water content. Knocking down *desat1* expression in all relevant tissues (including the fat body) induced very similar effects: the reduction of both desiccation resistance and the ratio of desaturated: linear alkanes. However, selective targeting of *desat1* expression in oenocytes strongly decreased the ratio of desaturated:linear alkanes, but did not affect desiccation resistance or water content. These data suggest that the relative increase of internal water content, but not the decreased ratio of desaturated: linear alkanes, is involved in reduced desiccation resistance. We also found evidence that a higher amount of internal water may not provide a decisive advantage in resisting desiccation: both *desat1* fat-body targeted genotypes—[RC-Gal4 × IR] and [6908-Gal4 × IR]—showed a lower LT50 but no slope change, suggesting that the earlier mean age of death shown by these transgenic flies may have been due to faster water loss compared to flies with a relatively lower water content.

It is possible to interpret these complex data in terms of insect physiology. One of the main functions of the fat body is to store and release energy, whereas oenocytes regulate lipid metabolism (*Ferveur, 1997*; *Gutierrez et al., 2007*). Energy is stored in adipocyte fat-body cells in the form of fatty acids, glycerol, triglycerides, and glycogen, which is stored in a bulky hydrated form (*Arrese & Soulages, 2010*). The amount of glycogen, which is normally lower than that of fat, can fluctuate depending on locomotor activity and environmental conditions (*Lorenz & Anand, 2004*). For example, if the insect is subject to freezing, low humidity or diapause, the fat body accumulates fat and eliminates water through the activity-dependent regulation of aquaporin water channel genes (*Izumi, Sonoda & Tsumuki, 2007*; *Liu et al., 2011*; *Sinclair & Renault, 2010*). We propose that in our *desat1* fat body-targeted flies the capacity to store lipids was affected. This would explain why, under our desiccation challenge, fat body knock-down transgenic flies lost water more rapidly and died earlier than controls.

Further insight into the link between desiccation resistance and CH levels can be found from examining the other mutants studied here. $rk^{1/4}$ mutant flies showed a highly increased $\sum$CH but a control D:L ratio of desaturated:linear alkanes. These flies died earlier than controls, showing that a high CH level, even combined with a control-like D:L ratio, cannot compensate for the underlying genetic defect. We suspect that the defective sclerotization process observed in the *rk* mutant affects the permeability of the adult cuticle: this could enhance both CH trafficking and water loss (*Flaven-Pouchon et al., 2016*; *Gibbs, 1998*; *Moussian, 2010*). Conversely, *rk*/CyO females with a single *rk* mutant allele and a control-like CH profile showed a lower LT50 and a shallower death slope than controls (this was not due to the CyO marker associated with the SM2 balancer). The reduction in the RK gene product may lead to a slowing down of water loss by changing the ultrastructure of the cuticle or of the spiracles (*Chown, 2002*; *Moussian, 2010*). The clear sex differences in desiccation resistance may also be related to differences in CH levels: mature females

express longer chain CHs than mature males (*Antony & Jallon, 1982*; *Gibbs, Chippindale & Rose, 1997*). This may explain why immature flies carrying CHs with longer carbon chains and higher number of double-bonds survived longer than mature flies under dry humidity condition (Fig. S1). However, no change in desiccation resistance was induced by the replacement of an alkene (7,11 HD) by a closely related isomer (5,9 HD) in Z30 mature females, which showed similar desiccation responses to Di2 females (Fig. 10).

## CONCLUSION

Our three-pronged approach to desiccation resistance and its underlying genetic and phenotypic components—selection, CH analysis and measurements of water content—provides insight into this fundamental aspect of the ecological physiology of insects. Our data suggest that desiccation resistance is not a simple phenotype: increased and decreased resistance depended on different hierarchies of physiological factors. There was no consistent relationship between increased resistance and various measures of cuticular hydrocarbon composition. Dessication resistance cannot be easily narrowed down to one or two variables, as seen by the different cuticular hydrocarbon composition of the strains studied here. Surprisingly, even the ability to retain a relatively high proportion of water was not related to desiccation resistance—transgenic flies with the highest proportion of water ([RC-Gal4 × IR] and [6908-Gal4 × IR]) showed the lowest levels of resistance. These two genotypes also showed reduced D:L (alkenes:alkanes) ratios underlining that a decline in desiccation resistance may occur in flies combining a high water content and low alkene levels. Flies in which these characters were dissociated showed no major change in desiccation resistance. All flies that died rapidly showed a very reduced LT50, but no change in their lethality slope: this suggests that early death equally affected all individuals of a given genotype.

Our study has revealed an intricate and non-linear relationship between desiccation resistance, CH profile and internal water content in *D. melanogaster* flies. These three phenotypes, which might be expected to show a simple relationship, turn out to have complex physiological and genetic links. While desiccation resistance and the proportion of desaturated CHs were tightly linked with both measures rapidly increasing after selection and persisting long after selection had been relaxed, a high water content negatively affected resistance, especially in association with a low level of desaturated CHs.

### Funding

This work was partly supported by the Centre National de la Recherche Scientifique (INSB), the Burgundy Regional Council (PARI 2014), the Université de Bourgogne and the CONICYT (MEC 80140013). There was no external additional funding received for this study. The funders had no role in study design, data collection and analysis, decision to publish, or preparation of the manuscript.

## Grant Disclosures

The following grant information was disclosed by the authors:
Centre National de la Recherche Scientifique (INSB).
Burgundy Regional Council: PARI 2014.
Université de Bourgogne and the CONICYT: MEC 80140013.

## Competing Interests

The authors declare there are no competing interests.

## Author Contributions

- Jean-Francois Ferveur conceived and designed the experiments, performed the experiments, analyzed the data, contributed reagents/materials/analysis tools, wrote the paper.
- Jérôme Cortot and Karen Rihani performed the experiments.
- Matthew Cobb wrote the paper, reviewed drafts of the paper.
- Claude Everaerts analyzed the data, wrote the paper, prepared figures and/or tables.

## Data Availability

  The raw data is provided in the Supplemental Files.

## Supplemental Information

Supplemental information for this article can be found online at http://dx.doi.org/10.7717/peerj.4318#supplemental-information.

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
