# Peer review of "Desiccation resistance: effect of cuticular hydrocarbons and water content in Drosophila melanogaster adults"

_PeerJ, doi:10.7717/peerj.4318_

## Round 0.1 · original submission · Major Revisions

Please answer the reviewer's concerns when re-submitting your article.

Reviewer 1 ·

Basic reporting

In this work, Ferveur and collaborators study desiccation resistance in Drosophila by analyzing cuticular hydrocarbon composition and its relationship with water content and with survival. For this, the authors compared wild type strains, artificially selected desiccation resistance flies, transgenic lines with reduced desaturase expression, which should affect cuticular hydrocarbon composition and hence desiccation resistance, and mutant flies with reduced cuticular hydrocarbon levels.

Experimental design

The authors clearly explain how they carried out their experiments, and include important controls like differences between males and females, and they discard the possibility of unavertedly selecting other traits that would ameliorate desiccation resistance like seize, and biases for the generation of males versus females in the offspring of different genetic backgrounds.

Validity of the findings

The conclusions of this work are valid, limited to and supported by the data provided. The authors show that desiccation resistance can be acquired and fixed in a population after only few generations, which persist without further selection through several generations. The work highlights the complex relationship between desiccation survival and the variables measured in the study. According to the data, it seems that the resistance cannot be easily narrowed down to one or two variables, and this is already apparent in the different cuticular hybrocarbon composition of different strains, for instance, Zimbabwe and Dijon2000.

Reviewer 2 ·

Basic reporting

The manuscript includes sufficient background. It shows clearly the context.
The structure of the article is according to the required format by PeerJ.
Figures are appropriately labelled and described.

Experimental design

Although the research question is well defined and relevant, the results found seem be very preliminary. The experimental design incorporates many variables so the results are very difficult to interpret and to make precise conclusions.

Validity of the findings

The representation and statistic analysis of the data are appropriate. The conclusions are very general and are not completely consistent with the results found.

Additional comments

The manuscript entitled “ Desiccation resistance: effect of cuticular hydrocarbons and water content in Drosophila melanogaster adults” by Ferveur et al. addresses a very interesting topic.
The authors studied the possible relationship between desiccation resistance, cuticular hydrocarbons (CHs) content and relative water content. Ferveur et al. found that an increased proportion of desaturated cuticular hydrocarbons, but not the total amount, is associated with an increased desiccation resistance. Also, authors found that transgenic flies with a high proportion of water showed the lowest levels of resistance. They conclude that desiccation resistance and the proportion of desaturated CHs are tightly linked while a high water content negatively affect desiccation resistance, at least when a low level of desaturated CHs are presented.

Although this work provides some interesting data, I consider that the results are very preliminary for publication leading to complex and confusing interpretations. For example, they found that lines 77S shown increased desiccation resistance respect to the control, it increased resistance was no significantly different between these lines but the ratio desaturated CHs: linear saturated CHs (D:L) was significantly different. Authors say that these results suggest that CH composition and desiccation resistance are not identical (manuscript lines 235-236) however they conclude that increased resistance was linked with increased ratios D:L (manuscript lines 382-383). In my opinion, the results do not show a consistent relationship between these two characteristics (desiccation resistance and ratio D:L).

In experiments using the UAS-desat1-RNAi transgene, authors said: “Knocking down desat1 in all relevant tissues induced a reduction of both desiccation resistance and the ratio D:L. However, selective targeting of desat1 expression in oenocytes strongly decreased the ratio D:L, but did not affect desiccation resistance or water content” (lines 336-340). Again, these observations are not consistent with what the authors propose regarding to a relationship between desiccation resistance and the ratio D:L. The results regarding water content and desiccation resistance are not convincing enough.

In experiments using RNAi lines it is desirable to use more than one RNAi line to avoid off-target effects and include some RNAi control line like GFP RNAi.


Minor comment:

In manuscript line 119. Change “homozygous” to “heteroallelic”.

---

## Round 0.2 · accepted · Accept

Congratulations! Your article is now acceptable for publication.

Reviewer 1 ·

Basic reporting

The authors have modified the text in a way that further strengthens its readability. They explain what their results imply regarding desiccation tolerance in flies, and they clarify the use of their fly lines according to findings reported in previous publications.

Experimental design

No comments.

Validity of the findings

No comments.

Reviewer 2 ·

Basic reporting

No comments

Experimental design

No comments

Validity of the findings

No comments

Additional comments

The authors have adequately modified the conclusions doing these more consistent with the results found. Although I consider that the results are somewhat preliminary, these are interesting and provide of information for future studies about the relationship of desiccation resistance with cuticular hydrocarbons and water content in insects.